# Adaptive CFAR Method for SAR Ship Detection Using Intensity and Texture Feature Fusion Attention Contrast Mechanism

**DOI:** 10.3390/s22218116

**Published:** 2022-10-23

**Authors:** Nana Li, Xueli Pan, Lixia Yang, Zhixiang Huang, Zhenhua Wu, Guoqing Zheng

**Affiliations:** 1Information Materials and Intelligent Sensing Laboratory of Anhui Province, Anhui University, Hefei 230601, China; 2East China Institute of Photo-Electron ICs, Suzhou 215163, China

**Keywords:** attention contrast mechanism, intensity dissimilarity, texture feature difference, generalized Gamma distribution (GΓD), CFAR ship target detection

## Abstract

Due to the complexity of sea surface environments, such as speckles and side lobes of ships, ship wake, etc., the detection of ship targets in synthetic aperture radar (SAR) images is still confronted with enormous challenges, especially for small ship targets. Aiming at the key problem of ship target detection in the complex environments, the article proposes a constant false alarm rate (CFAR) algorithm for SAR ship target detection based on the attention contrast mechanism of intensity and texture feature fusion. First of all, the local feature attention contrast enhancement is performed based on the intensity dissimilarity and the texture feature difference described by local binary pattern (LBP) between ship targets and sea clutter, so as to realize the target enhancement and background suppression. Furthermore, the adaptive CFAR ship target detection method based on generalized Gamma distribution (GΓD) which can fit the clutter well by the goodness-of-fit analyses is carried out. Finally, the public datasets HRSID and LS-SSDD-v1.0 are used to verify the effectiveness of the proposed detection method. A large number of experimental results show that the proposed method can suppress clutter background and speckle noise and improve the target-to-clutter rate (TCR) significantly, and has the relative high detection rate and low false alarm rate in the complex background and multi-target marine environments.

## 1. Introduction

Synthetic aperture radar (SAR) is an active microwave remote sensing device, which can acquire high-resolution images and work effectively in all weather and all day conditions. High-resolution SAR has a wide range of applications not only in military fields but also in civil fields, such as maritime traffic control and fishery management [1], and SAR technique has been widely used in sea surface surveillance in recent years. Therefore, it is of great significance to study how to accurately detect ship targets in SAR images.

The detection algorithm with the constant false alarm rate (CFAR) is widely used based on the statistical models of sea clutter, and two of typical detection methods are two-parameter CFAR and cell-average CFAR (CA-CFAR) based on a Gaussian model [2,3]. The outliers-robust CFAR detector was designed based on Gaussian clutter by the truncated-maximum-likelihood estimator and strong outliers could be removed so that the detection rate was improved in outliers-contaminated environment [4]. However, the motion of sea surface results in the difficulty of statistical modeling, and the statistical behaviors cannot be depicted by the Gaussian model. In recent years, a series of sea clutter distribution models were provided, mainly including Weibull distribution, Log-normal distribution, K distribution, Alpha-stable distribution, Gamma distribution, generalized Gamma distribution (GΓD), etc. [5,6,7,8,9,10,11,12,13,14]. Meanwhile, researchers proposed corresponding detection algorithms to realize the precise detection of ship targets. Farah et al. [15] presented a fast CFAR detection algorithm using GΓD model in SAR images with multiple ship targets. Considering the influence of ship targets for statistical modeling in the crowded harbors and busy shipping lines, Ai et al. [16,17] proposed the new two parameter CFAR detector in Log-normal clutter, and the method adopted the clutter truncation processing. In addition, the bilateral-trimmed-statistics CFAR detector with a closed-form solution was designed in multiple-target circumstances, and the outliers with high and low intensity are removed by bilateral-thresholds to improve the detection performance [18]. Pan et al. [19] proposed a CFAR ship detection method based on generalized K distribution, which removed outliers by the fuzzy statistical theory to obtain the precise sea clutter model and good detection performance in interference scenes. In order to improve the detection performance of ship targets with different scales in multi-targets scenarios, Dai et al. [20] provided the improved CFAR algorithm based on object proposals, which can not only obtain the precise position of the ship targets, but also improve the target detection rate. Leng et al. [21] comprehensively considered the intensity and spatial distribution of pixels and provided a bilateral CFAR algorithm, which utilized kernel density estimation (KDE) to extract spatial feature and performed CFAR detection by combining spatial distribution and intensity distribution to obtain more accurate spatial distribution characteristics and detection accuracy. In order to alleviate false positive from inherent speckle, discernible sea clutter, etc., Pappas et al. [22] provided a superpixel-level CFAR detection method based on the statistical model distribution, which can be applied in the previously models.

Due to the complexity of sea surface environments in SAR images, such as the effect of speckle noise and side lobes of ships, etc., it is still difficult to distinguish ship targets from sea clutter background by using the CFAR detection method based on a grayscale statistical feature in SAR images, especially small ships. In recent years, the attention contrast enhancement theory has been gradually applied to the object detection due to the great application potential for the ship extraction, and scholars have successively proposed ship target detection algorithms for improving the detection performance of ship targets. In order to reduce the influence of side lobes in SAR images on ship target detection, the fast multi-scale patch-based contrast measure (FMPCM) method presented could realize the target enhancement and mitigate the side lobes effect for the detection performance [23]. In addition, Wang et al. [24] introduced the multi-scale variance weighted information entropy (MVWIE) to measure the local dissimilarity between target and its surrounding background in SAR images with complex backgrounds. Considering the complexity of clutter features in high-resolution SAR images, Sun et al. [25] proposed a two-step ship target detection method based on coarse-to-fine mechanism, which can enhance ship targets based on the gravity field method. In view of the influence of speckle noise in sea clutter, Qian et al. [26] proposed a SAR ship target detection method based on multi-scale saliency analysis in frequency domain, which can achieve accurate target detection through frequency-domain multi-scale analysis and saliency information fusion, and has the advantage of low computational cost. Li et al. [27] proposed a contrast-based superpixel target detection method which performs target superpixel enhancement processing based on weighted information entropy (WIE) according to the difference of gray intensity between target superpixels and clutter superpixels. From the perspective of information geometry, Yang et al. [28] put forward a curvature-based saliency method based on information geometry theory and Riemannian manifold to improve the detection capability for ships. Wang et al. [29] proposed a novel unsupervised ship detection method using multi-scale saliency and complex signal kurtosis (MSS-CKS), which mainly includes two stages of extracting ship target candidate areas and discriminating true ship targets. The method uses a visual attention model to speed up the acquisition of target regions of interest in SAR images, and combines complex information to achieve efficient and real-time detection of ship targets. Because the structure and shape of the ship target are complex and the resolution cells contain weak echoes, the target detection performance is extremely challenging. Wang et al. [30] provided a high-resolution SAR ship target detection method, which was mainly based on hierarchical sparse model (HSM) of random forest to achieve rough detection of ship targets, and then used a contour saliency model (CSM) with a dynamic constant false alarm rate to achieve accurate detection of ship target contours. The ship target detection method using frequency enhanced maximally stable extreme region (MSER) was presented [31], which selected candidate ship targets by constructing the frequency domain saliency MSER, and utilized the CFAR method based on statistical distribution characteristics to complete accurately detection of targets. A superpixel-based local contrast measure (SLCM) method for ship detection has been provided, which can effectively avoid missing small targets and false alarms of strong noise points [32]. Due to the non-uniformity and nonstationarity of sea clutter distribution, ship target detection performance in the complex, and changeable sea clutter environment being limited, Zhang et al. [33] proposed a ship detection method based on information geometry (IG) theory by analyzing the statistical manifold and the geometric structure, which can realize the significant representation of ship targets and the suppression of clutter background.

In addition, with the development of deep learning in computer vision field, scholars have applied deep learning methods to the SAR ship detection, most common detection models contain single stage and two stage detectors. The You Only Look Once (YOLO) algorithm is a typical single-stage detection model, and scholars have successively proposed a series of improved algorithms based on the YOLO framework, such as You Only Look Once version 2 (YOLOv2) [34], H-YOLO [35], N-YOLO [36], You Only Look Once version 5 (YOLOv5) [37], Duplicate Bilateral YOLO (DB-YOLO) [38], etc. The region-based convolutional neural network (R-CNN) is a typical two-stage network model, which has been explored in the ship target detection task. In order to improve the detection accuracy, a large number of algorithms based on improved faster R-CNN have been provided [39,40,41]. Compared with the single-stage detection method, the two-stage method can often achieve higher detection accuracy, but its detection calculation speed is relatively slower. However, the deep learning detection method is data-driven, and the high-precision detection model needs vast amounts of data. Due to the limited amount of SAR data in the actual scenes, it is difficult to guarantee the detection performance in different SAR scenes.

At present, the conventional CFAR detection method may have the risk of mismatched models and low detection rate for small ships, and the deep learning detection method is the lack of real data for different scenarios. The detection performance of ship targets in complex marine environments still needs to be improved, and the further research of attention contrast enhancement is still essential and has the great prospect and wide applications, especially for the detection of small ship targets which seriously threatens the maritime safety. Aiming at the problem of ship target detection in complex SAR environments, especially for small ships, the adaptive CFAR algorithm in SAR images is designed based on the attention contrast mechanism in this article. In order to improve the detection performance, the attention contrast enhancement is processed based on the differences of the intensity feature and texture feature described by the non-uniform LBP between ship target and background in SAR images, and then CFAR ship target detection based on the excellent goodness-of-fit of statistical model is performed based on the attention contrast map.

The remainder of this article is organized as follows: Section 2 elaborates on the architecture of the proposed detection method of ship targets based on the intensity and texture features. Section 3 gives the experimental analyses of the proposed method and other benchmark methods, and discusses the effectiveness of the proposed method in different scenes. Section 4 summarizes the conclusions.

## 2. Detection Methodology

The sea clutter environment is complex and changeable, and there are generally obvious speckle characteristics in SAR images, and small ship targets have weak scattering and are easily hidden in the clutter so that the target detection rate decreases. Therefore, this article proposes an adaptive CFAR detection method for ship targets based on the local attention contrast enhancement. Because the use of a single intensity feature in a single-channel SAR image cannot characterize the local area information well, the texture feature can provide the spatial scattering distribution information of a ship due to the difference of scatterings. For this reason, this article combines the attention contrast difference of intensity and texture features in local regions to enhance feature representation of ship targets. Figure 1 gives the specific flow chart of the proposed detection method.

### 2.1. Contrast Enhancement Mechanism Based on Intensity Feature

The intensity of ships and sea clutter background is different, in order to describe the local difference of intensity feature, and considering the influence of targets distribution and the side lobes of ships for the detection performance, this article selects a local three-layer sliding window processing with a diagonal-oblique diagonal structure, as shown in Figure 2. The window is mainly divided into three layers, namely target window T, protection window P, and four-neighborhood background windows B1, B2, B3, and B4 on the diagonal-oblique diagonal regions. The window size can be selected according to the ship size.

Usually, the intensity of ship target pixels is stronger than the sea clutter background. The more similar the intensity characteristics of two local area blocks, the more likely the two areas belong to the same type (ship target or background). Based on these two facts, the intensity dissimilarity between the ship target block and its surrounding background blocks in the local sliding window area, as shown in Figure 2, is defined as
(1)$$\begin{array}{c} Lg_{k}=\mathrm{sgn}\left(\mu_{T}-\mu_{B_{k}}\right)\cdot \left(1-\frac{2\mu_{T}\mu_{B_{k}}}{\mu_{T}^{2}+\mu_{B_{k}}^{2}}\right)=\mathrm{sgn}\left(\mu_{T}-\mu_{B_{k}}\right)\cdot \left(\frac{{\left(\mu_{T}-\mu_{B_{k}}\right)}^{2}}{\mu_{T}^{2}+\mu_{B_{k}}^{2}}\right) \end{array}$$
where $\mu_{T}$ represents the mean intensity of the target block T in the local window area, $\mu_{B_{k}},k=1,2,3,4$ represents the mean intensity of the four surrounding background blocks, and symbol sgn is the sign function.

As shown in Equation (Equation 1), there are the following characteristics:When $\mu_{T}>\mu_{B_{k}}$, $\mathrm{sgn}(\mu_{T}-\mu_{B_{k}})=1$, it indicates that the intensity of the central target block T is stronger than the *k*-th background block in the local sliding window area, which means that the central target block T may be a candidate ship target.When $\mu_{T}<\mu_{B_{k}}$, $\mathrm{sgn}(\mu_{T}-\mu_{B_{k}})=-1$, it indicates that the intensity feature of the *k*-th background block is stronger than that of the central target block, which means that the central target block T may be the candidate background.When $\mu_{T}=\mu_{B_{k}}$, $\mathrm{sgn}(\mu_{T}-\mu_{B_{k}})=0$, it indicates that the intensity feature of the target block and the background block are highly similar, which means that the central target block T may be the ship target candidate area or candidate background area.When the intensity characteristics of the central target block and its surrounding background blocks are more dissimilar, that is, the greater the mean intensity difference is, the larger the value of ${\left(\mu_{T}-\mu_{B_{k}}\right)}^{2}$/$\left(\mu_{T}^{2}+\mu_{B_{k}}^{2}\right)$ is, and the value range is [0,1).

In order to achieve significant enhancement for ship targets, when the central target block T in the local sliding window area is the ship target candidate area, it is necessary to perform intensity feature attention enhancement processing on the central target block. Conversely, when the central target block T is the background candidate, the intensity suppression operation needs to be performed. Therefore, we combine the intensity feature of the central target block and the intensity dissimilarity between the ship target and surrounding backgrounds to obtain the local intensity feature contrast, and the contrast metric $C_{g}$ can be calculated as follows: (2)$$C_{g}\left(p,q\right)=\max\left\{e^{Lg_{k}}\cdot \frac{1}{N}\sum_{j=1}^{N}I_{T}^{j}\right\},k=1,2,3,4$$
where (p,q) is the center coordinate of the sliding window target block T, $I_{T}^{j}$ represents the *j*-th maximal intensity value of the target area T, and *N* is the number of maximal intensity values considered. When the average intensity of the central target block is greater than the average intensity of the background block, $e^{Lg_{k}}\in \left(1,e\right)$, so the ship targets can be enhanced. When the average intensity of the central target block is smaller than the average intensity of the background block, $e^{Lg_{k}}\in \left(e^{-1},1\right)$, the background pixel has a significant suppression effect. When the average intensity of the central target block is equal to the average intensity of the background block, $e^{Lg_{k}}=1$, it is unable to judge whether the central target block belongs to the ship target candidate area or the background candidate area, so the pixel of central target block is not enhanced.

### 2.2. Contrast Enhancement Mechanism Based on Texture Feature

The efficient detection of ship targets in SAR images is not easy to achieve only by the intensity feature, especially for the detection of small ship targets. In order to more accurately extract the ship targets from the complex sea clutter background, the attention contrast enhancement metric combines the texture difference between ship targets and clutter background in this article.

Local binary pattern (LBP) is mainly used to describe the local texture feature of the image, which reflects the relationship between any pixel in the image and its local neighborhood pixels. The traditional LBP mainly uses the intensity value of the central pixel as the threshold to compare the remaining pixels in the window, and characterizes the texture structure using binary encoding. The LBP value can be calculated by
(3)$$\begin{array}{c} LBP_{P}=\sum_{i=0}^{P-1}S_{LBP}\left(x_{i}-x_{c}\right)\cdot 2^{i}\\ s.t.S_{LBP}\left(t\right)\left\{\begin{array}{c} 0,t<0\\ 1,t\geq 0 \end{array}\right. \end{array}$$
where $x_{c}$ is the intensity value for the center pixel, $x_{i},i=0,1,...,P-1$ is the intensity value of *i*-th pixel in the local neighborhood, and *P* represents the number of neighbor pixels.

Therefore, stimulated by LBP, in order to adapt to the SAR image texture feature solution, we use the intensity differences between the central pixel (m,n) and all the pixels in its surrounding local neighborhood to reflect the texture characteristic change in the local region, and extracting the texture measure value at pixel (m,n) is denoted as
(4)$$\begin{array}{c} G_{tex}\left(m,n\right)={∥g_{tex}\left(m,n\right)∥}_{2}\\ s.t.g_{tex}\left(m,n\right)=\left(I_{0}-I\left(m,n\right),I_{1}-I\left(m,n\right),\ldots ,I_{L^{2}-2}-I\left(m,n\right)\right) \end{array}$$
where $g_{tex}\left(m,n\right)$ represents the feature vector of intensity differences between the pixel point (m,n) and its local neighborhood pixels, ${∥\cdot ∥}_{2}$ represents the 2-norm, and I(m,n) is the intensity value of pixel (m,n) in the original SAR image. $I_{j},j=0,1,...,L^{2}-2$ is the intensity value of *j*-th pixel in the L×L local neighborhood, and *L* represents the size of the local neighborhood window, which is generally an odd integer greater than or equal to 3. In general, the LBP mode of 3×3 neighborhood window is a uniform mode, but as the window size increases, the distribution of LBP mode will be uneven. The non-uniform LBP mode will provide more texture information in the window area composed of central pixel and its surrounding pixels [42], and *L* is chosen according to the small target size.

Due to the structure differences of a ship in different positions, the scattering is inhomogeneous after SAR imaging processing, and thus the scattering heterogeneity of a ship target will make the relative texture difference between the ship target and clutter background. That is, the texture of ship target is obviously stronger than clutter background. Therefore, the contrast enhancement metric using the difference of local texture feature between ship target and background is denoted as
(5)$$C_{te}\left(p,q\right)=\max\left\{{\bar{G}}_{T}\cdot \log_{2}\left({\bar{G}}_{T}/{\bar{G}}_{B_{k}}+1\right)\right\},k=1,2,3,4$$
where ${\bar{G}}_{T}$ and ${\bar{G}}_{B_{k}}$ represent the average texture measure value of the target block T and the average texture measure value of the surrounding background blocks in the local sliding window, respectively. When ${\bar{G}}_{T}>{\bar{G}}_{B_{k}}$, $\log_{2}\left({\bar{G}}_{T}/{\bar{G}}_{B_{k}}+1\right)>1$, which means that the texture of the target can be effectively enhanced. When ${\bar{G}}_{T}<{\bar{G}}_{B_{k}}$, $0<\log_{2}\left({\bar{G}}_{T}/{\bar{G}}_{B_{k}}+1\right)<1$, which means that it has a certain inhibitory effect on the texture feature of the central target block. When ${\bar{G}}_{T}={\bar{G}}_{B_{k}}$, $\log_{2}\left({\bar{G}}_{T}/{\bar{G}}_{B_{k}}+1\right)=1$, the texture enhancement is not performed on the center target block.

### 2.3. CFAR Target Detection Based on Clutter Statistics

Combining attention contrast enhancement metric based on intensity and texture features, the final contrast mechanism $C_{s}$ is obtained by the fusion processing, and calculated by
(6)$$C_{s}=C_{g}\times C_{te}$$

In order to adaptively extract ship targets from the background after the attention contrast enhancement processing, the adaptive CFAR detection method based on clutter statistical distribution model is carried out. The distribution model of attention contrast map with the superior fitting accuracy is selected according to the statistical characteristic analyses by a large number of different actual scenes. The analytical statistical models mainly include Gauss distribution model, Rayleigh distribution model, Weibull distribution model, Log-normal distribution model, Gamma distribution model, and GΓD model. In order to illustrate the goodness-of-fit of different models in this article, we give experimental results of the clutter background regions after the attention contrast enhancement processing from different SAR scenes, and the selected regions are the pure clutter regions C1, C2, C3, and C4 marked by the black dotted rectangle shown in Figure 3a–c. The region C1 is the relative uniform sea clutter background, the region C2 is the clutter area with obvious texture undulation, the region C3 represents the clutter background with strong speckle noise, and the region C4 stands for the non-uniform clutter background. The fitting results of different regions are shown in Figure 4a–d.

In order to quantitatively analyze the fitting performance of real data, the mean square difference (MSD) is used to evaluate the goodness-of-fit, and defined as
(7)$$D_{msd}=\frac{1}{N_{c}}\sum_{i=1}^{N_{c}}{\left(p_{t}\left(x_{i}\right)-p_{m}\left(x_{i}\right)\right)}^{2}$$
where $p_{t}\left(x_{i}\right)$ and $p_{m}\left(x_{i}\right)$ represent the theoretical probability density function (PDF) and histogram probability of measured data, respectively. $N_{c}$ is the length of the data sequence. The smaller the MSD value is, the better the clutter fitting effect will be.

It can be obviously seen from the clutter fitting results in Figure 4a–d that the fitting performance of GΓD model is better in the low amplitude ranges. The fitting accuracy of clutter tail is the key for achieving high-precision detection, and in order to compare the fitting accuracy of tail for various distribution models, Table 1 lists the fitting MSD values of different statistical models for the background tail, and it can be seen that the GΓD model has the smallest fitting error to the clutter tail.

Combining the clutter fitting curves of different statistical models and MSD values of the clutter tail goodness-of-fit, it is clearly verified the GΓD model has the best fitting effect for real data after the proposed attention contrast enhancement method. The CFAR ship target detection is performed based on GΓD [11,12], whose PDF can be written as
(8)$$p_{t}\left(x\right)=\frac{\left|v\right|k^{k}}{\delta \Gamma (k)}{\left(\frac{x}{\delta }\right)}^{kv-1}\exp\left\{-k{\left(\frac{x}{\delta }\right)}^{v}\right\},\delta ,\left|v\right|,k,x>0$$
where δ, *v*, and *k* are the scale parameter, power parameter, and shape parameter, respectively, and Γ(·) represents the Gamma function.

The relationship between the detection threshold and probability of false alarm (PFA) is
(9)$$Th=\left\{\begin{array}{c} \delta {\left\{\frac{1}{k}Q_{Inv}\left(1-P_{fa},k\right)\right\}}^{\frac{1}{v}},v>0\\ \delta {\left\{\frac{1}{k}Q_{Inv}\left(P_{fa},k\right)\right\}}^{\frac{1}{v}},v<0 \end{array}\right.$$
where $Q_{Inv}\left(\cdot \right)$ represents the inverse incomplete gamma function, and $P_{fa}$ is the specified PFA.

## 3. Results and Discussion

This algorithm verification selects the high-resolution SAR images collected by TerraSAR-X satellite and Sentinel-1 satellite in the public datasets HRSID [43] and LS-SSDD-v1.0 [44]. The performance of the proposed method is verified from the aspects of target enhancement effect and target detection performance, the comparative analyses are made with CA-CFAR method [3], GΓD-CFAR method [11], new two parameter CFAR method based on Log-normal distribution (LN-2PCFAR) [17], FMPCM method [23], MVWIE method [24], and IG method [33].

### 3.1. Target Enhancement Performance Analyses

First of all, in order to verify the contrast enhancement effect of proposed method, especially the enhancement performance for small ship targets, we conducted a large number of test experiments using SAR images from the public datasets HRSID and LS-SSDD-v1.0. The article selects the SAR image with small ship targets to show the target enhancement effect as shown in Figure 5a, where the small target slice marked by the red dotted rectangle, and the small ships are marked as T1, T2, and T3 in Figure 5b. The proposed method is compared with FMPCM and MVWIE methods to verify the contrast enhancement performance.

Figure 6a–d show three-dimensional images of Figure 5b, the MVWIE method, the FMPCM method, and the proposed method, respectively. As shown in Figure 6a–d, the small target in the original SAR image has a low contrast with its surrounding clutter, and it is not easy to distinguish from the clutter background. After using the MVWIE method, FMPCM method, and our method, the contrast between ship targets and the clutter background can be improved. However, the suppression effect of MVWIE and FMPCM methods is relatively worse than proposed method, and there is still clutter remaining which will induce false alarms. After the attention contrast enhancement of the intensity and texture features, our proposed method has a better inhibition effect for the clutter background.

In order to further quantitatively illustrate the enhancement effect, the TCR is used and calculated as
(10)$$TCR=10\log_{10}\left(S_{t}/S_{c}\right)$$
where $S_{t}$ and $S_{c}$ represent the average power of the detected ship target pixels and surrounding clutter background pixels, respectively.

Table 2 presents the original TCR value and TCR values after target enhancement processing based on different enhancement algorithms. Compared with the MVWIE algorithm, the proposed method can enhance the TCR by 10∼13 dB, and compared with the FMPCM algorithm, the proposed method can enhance the TCR by 6∼8 dB. It can be seen that the proposed method in this article can significantly improve TCRs from Table 2 and Figure 6, which proves that the algorithm has a good target enhancement effect so that the detection rate of ships can be increased effectively.

### 3.2. Target Detection Performance Analyses

In order to validate the detection performance of our method for different scenarios, lots of target detection experiments are conducted using real SAR images from the public datasets HRSID and LS-SSDD-v1.0. This article chooses five different SAR images with multiple targets to show the detection effect of ship targets, and at the same time, the proposed method is compared with CA-CFAR method, GΓD-CFAR method, new LN-2PCFAR method, FMPCM method, MVWIE method, and IG method. In order to compare the detection performance of different algorithms, all ships can be detected for each experimental scenario with under the same PFA. Figure 7a, Figure 8a, Figure 9a, Figure 10a and Figure 11a present five original multi-target SAR images, respectively, and the real ship targets are marked with white rectangles.

Figure 7, Figure 8, Figure 9 and Figure 10 and Figure 11b–f show ship target detection results for CA-CFAR method, GΓD-CFAR method, new LN-2PCFAR method, MVWIE method, FMPCM method, IG method, and our proposed method in this article, respectively. Correctly detected targets, false alarms are marked as the red, green rectangles after the clustering processing. It can be seen from the all detection results that the CA-CFAR method, GΓD-CFAR method, and the new LN-2PCFAR method will cause more false alarms due to the influence of interference (speckles, ship wake, etc.). The MVWIE, FMPCM, and IG methods are not effective enough to suppress the interference, and there are still a few false alarms. As shown in Figure 7 and Figure 8, it can be seen that the MVWIE method, FMPCM method, and IG method will misjudge the strong interference points as the target pixels, and for the small target scenarios, it can be seen that the intensity difference between bright noise points and small ship targets is not obvious as shown in Figure 9, and the MVWIE method, FMPCM method, IG method, and the proposed method can enhance some obvious speckle noises while enhancing ship targets, thus causing false alarms. However, our proposed method can suppress the background so that the false alarm rate decreases as shown in Figure 8 and Figure 9. In addition, due to the discontinuity of scatterings for ship target, a ship may be mistaken as multiple ships as shown in Figure 8, and the proposed detection method can alleviate this phenomenon. It can be seen that the detection ability has the same level of efficiency for the FMPCM method, IG method, and our proposed method in a relatively uniform scene from Figure 10 and Figure 11. Compared with other methods, our proposed method has a better suppression effect on the side lobe generated by the strong scattering characteristics of ship target T0 marked in Figure 11a.

Furthermore, figure of merit (FoM) is used to quantitatively evaluate the detection performance and was denoted as
(11)$$FoM=\frac{N_{d}}{N_{f}+N_{r}}$$
where $N_{d}$ represents the number of correctly detected ship targets, $N_{f}$ is the number of false alarms, and $N_{r}$ represents the number of real targets in a SAR image. As shown in Equation (Equation 11), the larger the value of FoM, the higher the target detection rate.

Table 3 lists the FoM values calculated by different methods. As shown in Table 3, the CA-CFAR method, GΓD-CFAR, and new LN-2PCFAR methods will produce false alarms in complex environments. Since the MVWIE and FMPCM methods use single feature contrast metric, the suppression effect of strong speckle noises may not be good enough, which will result in a few false alarms. By constructing statistical manifold space, the IG method realizes the difference representation between the target and the background area, which improves the target detection to some extent, but there is still a small amount of false alarms when there is obvious speckle noise. However, the algorithm in this article significantly improves TCRs and the probability of missed detection and false alarm is decreased after the attention contrast enhancement processing based on the intensity and texture feature differences. Therefore, the proposed method in this article improves the detection performance of ship targets.

## 4. Conclusions

In this article, the adaptive CFAR ship target detection method based on an attention contrast enhancement mechanism is presented, and the effectiveness of this method is verified using a lot of multi-target scene SAR images. The algorithm firstly combines the intensity feature of the target block in the local sliding window area and its intensity dissimilarity with different surrounding background blocks, and based on the texture structure described by non-uniform LBP, the texture difference between the target block and the clutter background block in the local window area is counted. Then, the target attention contrast enhancement is achieved by combining the intensity and texture feature differences. After the attention contrast enhancement processing, the optimal selection of the statistical distribution model is achieved through the goodness-of-fit, analyses, and the CFAR ship target detection is carried out based on the optimal GΓD model. Through a large number of experiments and comparative analyses of different methods, it is verified that the algorithm in this article can significantly improve the TCR. In addition, it has a better enhancement effect for ships and background suppression effect in a complex multi-target environment, reduces the false alarm rate caused by speckle, and significantly improves the detection performance of small ship targets. In the future, the more effective feature differences between ship targets and clutter background and fused mechanism will be further explored to improve the detection accuracy and sea clutter suppression ability in complex scenes. 

## Figures and Tables

**Figure 1 sensors-22-08116-f001:**
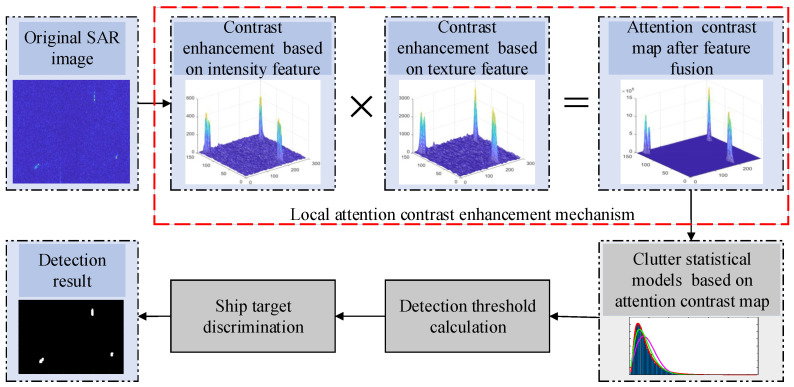
Flow chart of the proposed method.

**Figure 2 sensors-22-08116-f002:**
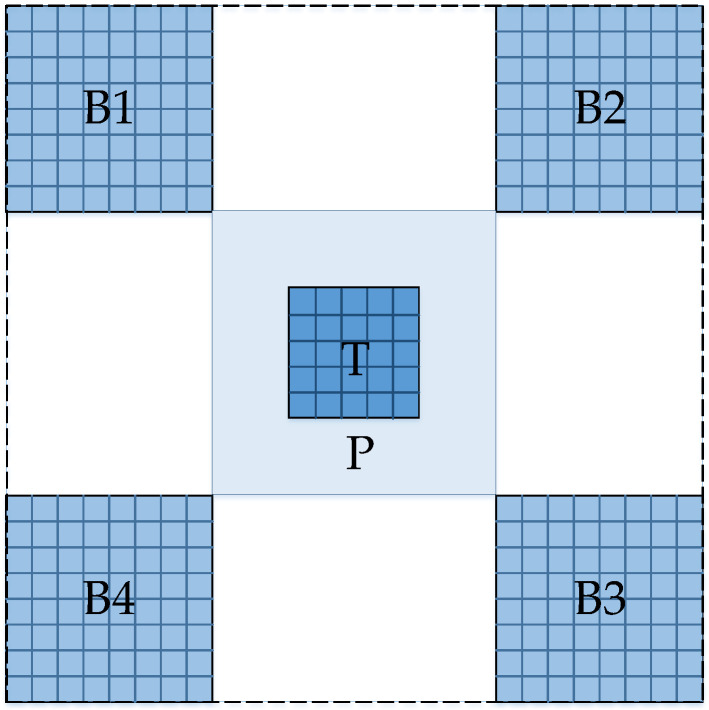
Structure diagram of the local sliding window.

**Figure 3 sensors-22-08116-f003:**
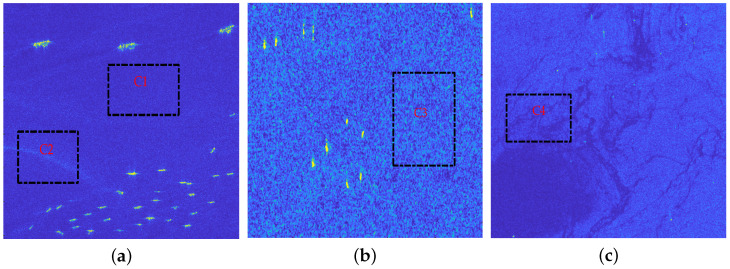
Real SAR scenes. (**a**) Scene 1; (**b**) Scene 2; (**c**) Scene 3. The pure clutter regions C1, C2, C3, and C4 are marked with black dotted rectangles.

**Figure 4 sensors-22-08116-f004:**
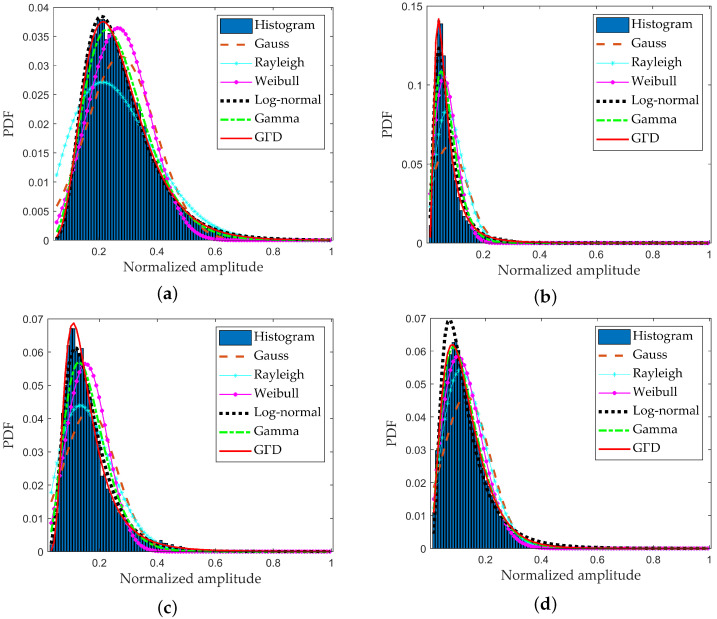
Fitting results of different regions. (**a**) fitting result of region C1; (**b**) fitting result of region C2; (**c**) fitting result of region C3; (**d**) fitting result of region C4.

**Figure 5 sensors-22-08116-f005:**
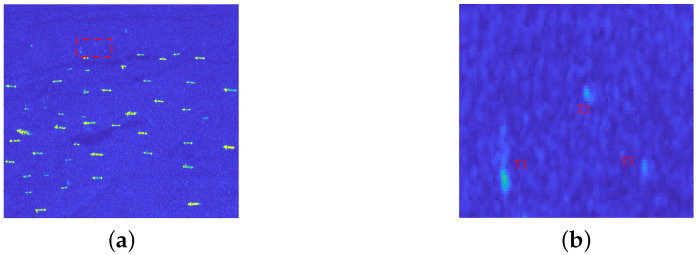
Scene 4 with small ship targets and small target slice. (**a**) scene 4; (**b**) small target slice marked by the red dotted rectangle in (**a**).

**Figure 6 sensors-22-08116-f006:**
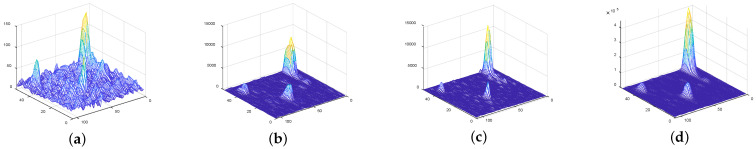
Target enhancement results. (**a**) original small target slices; (**b**) results of the MVWIE method; (**c**) results of the FMPCM method; (**d**) results of the proposed method.

**Figure 7 sensors-22-08116-f007:**
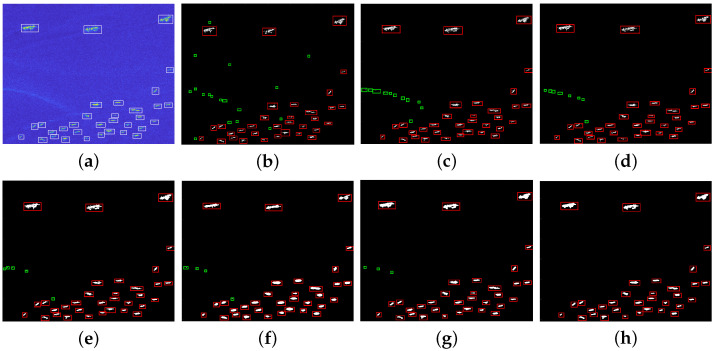
Detection results of different methods for scene 1. (**a**) original image; (**b**) CA-CFAR method; (**c**) GΓD-CFAR method; (**d**) new LN-2PCFAR method; (**e**) MVWIE method; (**f**) FMPCM method; (**g**) IG method; (**h**) proposed method.

**Figure 8 sensors-22-08116-f008:**
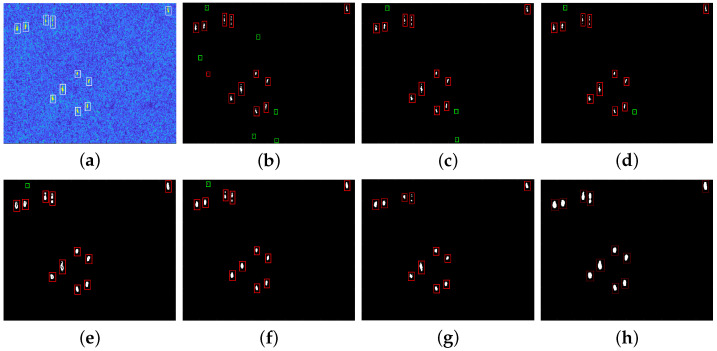
Detection results of different methods for scene 2. (**a**) original image; (**b**) CA-CFAR method; (**c**) GΓD-CFAR method; (**d**) New LN-2PCFAR method; (**e**) MVWIE method; (**f**) FMPCM method; (**g**) IG method; (**h**) proposed method.

**Figure 9 sensors-22-08116-f009:**
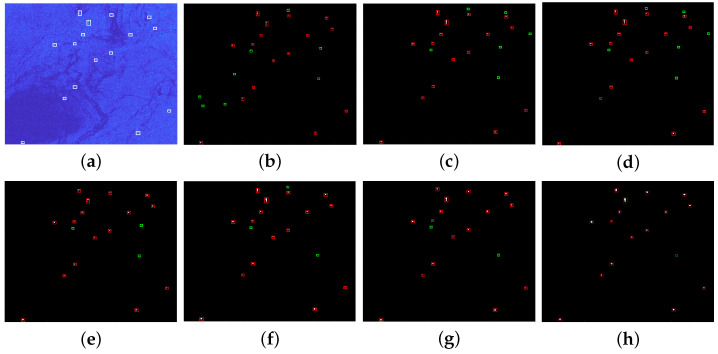
Detection results of different methods for scene 3. (**a**) original image; (**b**) CA-CFAR method; (**c**) GΓD-CFAR method; (**d**) new LN-2PCFAR method; (**e**) MVWIE method; (**f**) FMPCM method; (**g**) IG method; (**h**) proposed method.

**Figure 10 sensors-22-08116-f010:**
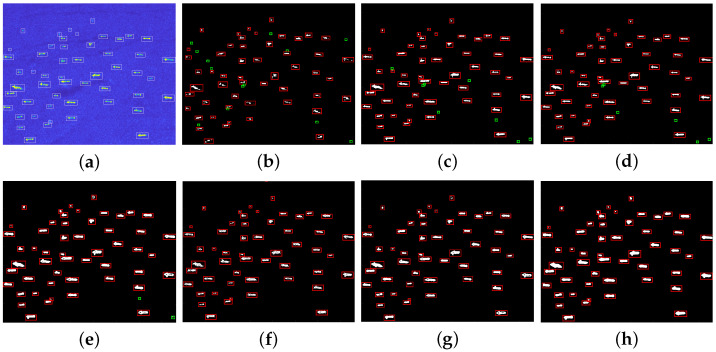
Detection results of different methods for scene 4. (**a**) original image; (**b**) CA-CFAR method; (**c**) GΓD-CFAR method; (**d**) new LN-2PCFAR method; (**e**) MVWIE method; (**f**) FMPCM method; (**g**) IG method; (**h**) proposed method.

**Figure 11 sensors-22-08116-f011:**
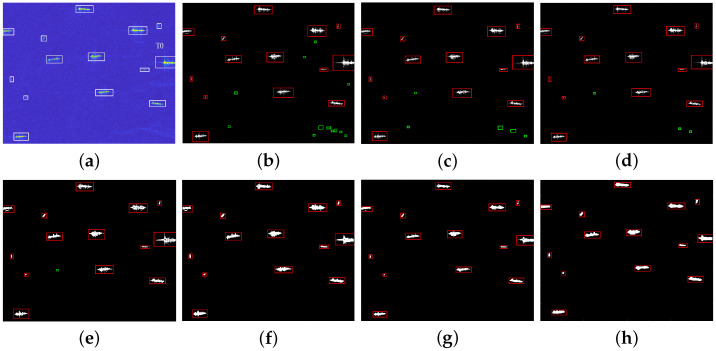
Detection results of different methods for scene 5. (**a**) original image; (**b**) CA-CFAR method; (**c**) GΓD-CFAR method; (**d**) new LN-2PCFAR method; (**e**) MVWIE method; (**f**) FMPCM method; (**g**) IG method; (**h**) proposed method.

**Table 1 sensors-22-08116-t001:** The fitting MSD values of different statistical models for clutter tail in different regions.

Distribution Models	Region C1	Region C2	Region C3	Region C4
Gauss	$7.4857\times 10^{-9}$	$4.5551\times 10^{-9}$	$1.8173\times 10^{-8}$	$2.2635\times 10^{-7}$
Rayleigh	$3.9911\times 10^{-9}$	$4.5551\times 10^{-9}$	$1.7713\times 10^{-8}$	$1.9565\times 10^{-7}$
Weibull	$7.5013\times 10^{-9}$	$4.5551\times 10^{-9}$	$1.8185\times 10^{-8}$	$2.4318\times 10^{-7}$
Gamma	$5.1426\times 10^{-9}$	$4.5531\times 10^{-9}$	$1.7336\times 10^{-8}$	$7.5082\times 10^{-8}$
Log-normal	$3.4524\times 10^{-9}$	$2.9926\times 10^{-9}$	$8.4293\times 10^{-9}$	$4.5704\times 10^{-8}$
GΓD	$1.7642\times 10^{-9}$	$9.2214\times 10^{-10}$	$8.0524\times 10^{-9}$	$3.3507\times 10^{-8}$

**Table 2 sensors-22-08116-t002:** TCR values calculated by different methods.

Targets	Original TCR/dB	MVWIE Method TCR/dB	FMPCM Method TCR/dB	Proposed Method TCR/dB
T1	14.5718	26.2192	32.8889	39.3563
T2	11.2389	22.3531	25.5922	33.6248
T3	10.2377	19.6078	22.8028	29.5424

**Table 3 sensors-22-08116-t003:** FoM values of different algorithms.

Scenes	$N_{r}$	CA-CFAR	GΓD-CFAR	New LN-2PCFAR	MVWIE	FMPCM	IG Method	Proposed Method
Scene1	33	0.6471	0.7500	0.8049	0.8684	0.8919	0.9167	1
Scene2	11	0.6471	0.7857	0.8462	0.9167	0.9167	1	1
Scene3	16	0.6667	0.7273	0.7273	0.8421	0.8421	0.8889	0.9412
Scene4	52	0.8000	0.8667	0.8966	0.9630	1	1	1
Scene5	15	0.5556	0.7143	0.8333	0.8824	0.9375	1	1

## Data Availability

Not applicable.

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
