# Peer review of "Adaptive CFAR Method for SAR Ship Detection Using Intensity and Texture Feature Fusion Attention Contrast Mechanism"

_sensors, 2022, doi:10.3390/s22218116_

Round 1

Reviewer 1 Report

The article proposes a constant false alarm rate (CFAR) algorithm for synthetic aperture radar (SAR) images to detect ship target based on the attention contrast mechanism of intensity and texture feature fusion. The topic is interesting and the paper is well organized. However, there are some major comments in the following points.

1- The title is general. It should be more specific and related to the methods used for detection and related to the detected objects.

 2- The paper needs more justifications and detailed about the results. For example. In Figure 8, why the detected objects of your method are thicker than the other methods?

3- The authors should add results on some other images with different and more complex backgrounds.

4- Compare your results with more recent methods.

5- More references of some recent methods relevant the topic should be reviewed and discuss their advantages and disadvantages.

6- The conclusions should be supported by the results, showing why and how it can significantly improve the TCR and its limitations as a future direction for researchers.

Reviewer 2 Report

The authors have presented a manuscript that addresses the very attractive problem of detecting ships at sea using SAR sensors, with a focus on detecting small ships in a group of many objects at sea, i.e., in a complex environment. The developed model has been tested with a large number of images from a public dataset. In this work, the performance of the algorithm is demonstrated by analysing four selected images from SAR, where the proposed method (based on the mechanism of attentional contrast enhancement) shows the best results - it is compared with three known approaches. The paper includes all the elements relevant to the demonstration of a scientific paper, except for the potential limitations of this method. Here we are most interested in how the proposed method performs in azimuth ambiguities; it appears that this aspect is not discussed or tested at all. We are also interested in the proportion of false-positive detections.

Round 2

Reviewer 1 Report

Thank you for improving the quality of the manuscript. The authors updated the manuscript according to the review comments.